# Extremely Rare Complications in Uniportal Spinal Endoscopy: A Systematic Review with Unique Case Analyses

**DOI:** 10.3390/jcm13061765

**Published:** 2024-03-19

**Authors:** Kajetan Łątka, Waldemar Kołodziej, Dawid Pawuś, Marek Waligóra, Jacek Trompeta, Tomasz Klepinowski, Piotr Lasowy, Masato Tanaka, Beata Łabuz-Roszak, Dariusz Łątka

**Affiliations:** 1Department of Neurosurgery, St. Hedwig’s Regional Specialist Hospital, 45-221 Opole, Poland; waldemar.kolodziej@usk.opole.pl (W.K.); piolasowy@gmail.com (P.L.); dariusz.latka@gmail.com (D.Ł.); 2Department of Neurosurgery, University Hospital, Institute of Medical Sciences, University of Opole, 45-040 Opole, Poland; 3Department of Neurology, St. Hedwig’s Regional Specialist Hospital, Institute of Medical Sciences, University of Opole, 45-040 Opole, Poland; beatamaria.pl@hoga.pl; 4Faculty of Electrical Engineering, Automatic Control and Informatics, Opole University of Technology, 45-758 Opole, Poland; dawid.pawus@o2.pl; 5Clinical Department of Diagnostic Imaging, Institute of Medical Sciences, University of Opole, 45-040 Opole, Poland; marekwaligoraluxmed@gmail.com; 6Department of Orthopedic Surgery Piekary Slaskie, 41-940 Piekary Śląskie, Poland; jacektrompeta@gmail.com; 7Department of Neurosurgery, Pomeranian Medical University Hospital No. 1, Unii Lubelskiej 1, 71-252 Szczecin, Poland; tomasz.klepinowski@pum.edu.pl; 8Department of Orthopedic Surgery, Okayama Rosai Hospital, Okayama 702-8055, Japan; tanaka0896@gmail.com

**Keywords:** endoscopic spine surgery, minimally invasive techniques, surgical complications, patient safety, learning curve, surgical outcomes, case reports, spinal disorders, surgical education, technological advancements

## Abstract

**Background:** Endoscopic spine surgery represents a significant advancement in the minimally invasive treatment of spinal disorders, promising reduced surgical invasiveness while aiming to maintain or improve clinical outcomes. This study undertakes a comprehensive review of the literature on endoscopic spine surgery, with a particular focus on cataloging and analyzing the range of complications, from common postoperative issues to more severe, casuistic outcomes like dural tears and nerve damage. **Methods:** Our methodology encompassed a detailed review of meta-analyses, prospective randomized trials, cohort studies, and case reports to capture a broad spectrum of complications associated with endoscopic spine techniques. The emphasis was on identifying both the frequency and severity of these complications to understand better the procedural risks. **Results:** The findings suggest that endoscopic spine surgery generally exhibits a lower complication rate compared to traditional surgical approaches. Nonetheless, the identification of specific, rare complications peculiar to endoscopic methods underscores the critical need for surgeons’ advanced skills, continuous learning, and awareness of potential risks. **Conclusions:** Recognizing and preparing for the potential complications associated with the rapid adoption of endoscopic techniques is paramount to ensuring patient safety and improving surgical outcomes in minimally invasive spine surgery.

## 1. Introduction

Spinal endoscopy represents a significant advancement in minimally invasive spine surgery, offering a less-invasive alternative to traditional surgical methods [1]. Its emergence has sparked considerable interest due to its potential to minimize surgical trauma, reduce recovery times, and decrease the risk of postoperative complications [2]. By allowing surgeons to perform complex spinal procedures through small incisions, endoscopic techniques disrupt less muscle and tissue, potentially leading to better patient outcomes [3]. Spinal endoscopy has found broad application primarily in the treatment of degenerative spine diseases, but also in managing inflammatory conditions [4] and even in oncological procedures [5].

The safety and efficacy of spinal endoscopy have been increasingly highlighted in contemporary research, suggesting a lower incidence of complications when compared to conventional spine surgery [6,7,8,9,10,11]. Despite these advantages, it is important to acknowledge that spinal endoscopy, like any surgical intervention, carries the risk of adverse outcomes. These can range from minor issues, such as postoperative discomfort, to more serious complications, including dural tears or nerve damage.

Given the growing adoption of endoscopic spine surgery, especially in outpatient settings [12], it is essential to thoroughly understand the potential complications associated with this approach. This knowledge is crucial for surgeons who are considering or beginning to implement these techniques in their practice.

This article aims to explore the complications linked with endoscopic spine surgery, focusing on both the commonly reported issues and exceptionally rare cases that have not been extensively documented in the literature. Recognizing that endoscopic procedures are generally considered safer with fewer complications does not negate the importance of being aware of these potential risks. Our goal is to review these complications, highlight case reports of unique incidents, and present three novel case descriptions of complications not previously reported. This is particularly relevant as spinal endoscopy and other minimally invasive techniques become increasingly prevalent in outpatient surgical centers. Raising awareness of rare complications will enable surgeons to better prepare for and manage these challenges, thereby enhancing patient care in the realm of minimally invasive spine surgery.

## 2. Materials and Methods

Literature Review Strategy: Our comprehensive literature review was centered on endoscopic spine surgery techniques, including PELD, TELD, IELD, microdiscectomy, and microendoscopic discectomy. We explored several key medical databases, such as PubMed, Cochrane, and Web of Science, to amass a diverse collection of pertinent literature. This review was conducted in alignment with the Preferred Reporting Items for Systematic Reviews and Meta-Analyses (PRISMA) statement, ensuring a structured and transparent approach to data collection and analysis.

Search Strategy: We employed a detailed search strategy using specific keywords like “endoscopic spine surgery”, “percutaneous endoscopic lumbar discectomy”, “transforaminal endoscopic lumbar discectomy”, “interlaminar endoscopic lumbar discectomy”, “microdiscectomy”, “microendoscopic discectomy”, and “complications”. Our goal was to unearth meta-analyses and literature reviews through which we could further identify prospective randomized trials and cohort studies, delving into individual case reports of complications.

Selection Criteria: Our focus was primarily on meta-analyses and literature reviews that underscored the frequency and nature of complications associated with endoscopic spine surgery. Initial screening of titles and abstracts led us to these comprehensive analyses, which were then meticulously reviewed to confirm relevance and distill essential data. Through these secondary sources, we pinpointed prospective randomized trials and cohort studies for further examination.

Inclusion Criteria: We included articles detailing complications in lumbar endoscopic discectomy, meta-analyses comparing endoscopic discectomy with microdiscectomy or other surgical techniques, and literature reviews concerning complications in endoscopic discectomy. This search was extended to identify individual case reports within prospective randomized trials and cohort studies, showcasing rare complications.

Exclusion Criteria: We excluded articles focused on thoracic and cervical endoscopy, biportal endoscopy, endoscopic decompression in stenosis, and stabilization using endoscopy, ensuring our review remained targeted towards lumbar endoscopic discectomy.

Data Analysis: Our analysis of the selected meta-analyses and literature reviews aimed to spotlight the most frequently reported complications across different endoscopic spine surgery modalities. We paid special attention to unique complications inherent to endoscopic procedures and sought patterns or specific risk factors contributing to these complications. After our initial exploration, we dove deeper into the literature, including prospective and cohort studies included in meta-analyses and case reports embedded within. This allowed us to uncover rare complications not widely reported, thereby enriching the existing body of knowledge on the subject with previously undocumented case studies of complications.

A statistical analysis was conducted to compare the occurrence of various complications between the groups of patients undergoing endoscopic procedures and the group of patients undergoing microdiscectomy. The complication rate was calculated as the number of complications divided by the total number of procedures performed in a given group, multiplied by 100%. To compare the frequency of the occurrence of individual complications between the groups, the chi-square test was employed. This test is suitable for assessing associations between categorical variables. The chi-square statistic was calculated based on the contingency table, which contained the number of cases for different categories of complications in both groups. The level of statistical significance was set at *p* < 0.05. A *p*-value less than 0.05 was considered statistically significant, indicating that there are significant differences in the occurrence of complications between the groups. All statistical analyses were performed using Matlab 2020b software (MathWorks, Natick, MA, USA).

## 3. Results

Our comprehensive literature search across multiple databases identified a total of 418 records. From these, we removed 203 duplicates, narrowing our pool to 215 records. These records were then screened based on title and abstract, which led to the exclusion of 36 records that were not in English, 65 records that included biportal studies, 15 records related to endoscopic fixation, and 19 records that were not specific to lumbar procedures (Figure 1).

Following the initial screening, 80 full-text articles were assessed for eligibility. At this stage, further exclusions were made based on specific criteria: 12 records were excluded because they did not pertain to fully endoscopic procedures, and 8 records were excluded due to their focus on stenosis rather than disc herniation or other relevant lumbar pathologies.

Our exhaustive search across multiple databases led us to identify 11 meta-analyses, 14 prospective randomized trials (RCTs), and 30 cohort studies, focusing on the complications associated with endoscopic discectomy in the lumbar spine. Upon reviewing the meta-analyses, it was observed that they incorporated 10 of the 14 RCTs and 25 of the 30 cohort studies we identified. This inclusion criterion ensured a comprehensive aggregation of data, providing a robust foundation for our analysis. Further exploration was dedicated to identifying case reports detailing complications, resulting in the discovery of three pertinent articles. In addition to the case reports, our search also unveiled two review articles addressing complications, which included descriptions of atypical complications. This further enriched our understanding of the diverse range of adverse outcomes associated with endoscopic spine surgery.

### 3.1. Complication Rates

In our analysis of the meta-analyses for complication-related insights, all studies conducted a systematic examination of complications. Eight of these studies calculated the incidence rates of complications, with only one making a distinction between the transforaminal and interlaminar approaches. Five meta-analyses categorized complications by type, enabling us to identify the most reported complications across the studies. Importantly, only three meta-analyses indicated a significant difference in the frequency of complications favoring endoscopy [9,10,11], while the remaining studies did not demonstrate statistically significant differences (Table 1).

### 3.2. Common Complications

Subsequently, we conducted an analysis of all identified randomized and cohort studies to search for descriptions of individual case-specific complications that occurred. Notably, in 13 of the 14 RCTs, complications were discussed, and the frequency of their occurrence was reported. This thorough examination allowed us to categorize the complications commonly associated with endoscopic spine surgery. Among the most frequently reported complications were sensory disturbances, incomplete herniation removal, durotomy, nerve root damage, hematomas, discitis, and instability (Table 2).

Upon conducting a statistical analysis comparing complications from all randomized controlled trials (RCTs) we examined (Table 3), we arrived at the following results: The comparative analysis of complication rates among different lumbar discectomy techniques—specifically transforaminal endoscopic lumbar discectomy (TELD), interlaminar endoscopic lumbar discectomy (IELD), and microscopic lumbar discectomy (MLD), with percutaneous endoscopic lumbar discectomy (PELD) encompassing both TELD and IELD techniques as one group—revealed distinct statistical significance in safety and complication rates. Notably, TELD was found to be significantly safer than IELD, as indicated by a *p* value of less than 0.0001. In contrast, no significant difference in safety was observed between TELD and MLD, as evidenced by a *p* value of 0.18, suggesting that these techniques have comparable safety profiles. The comparison between IELD and MLD resulted in a *p* value of 0.04, indicating that MLD is statistically safer than IELD. Furthermore, when comparing both endoscopic approaches as a single group under PELD, it demonstrated a higher safety profile compared to MLD, with a *p* value of 0.0092, underlining PELD as the safer option among the techniques evaluated. This inclusion of both TELD and IELD under the umbrella of PELD highlights the overall safety benefits of endoscopic approaches when considering the risk of complications (Table 4).

Despite the lower complication rate associated with endoscopy, it’s important to note that the reoperation rate for MLD is statistically lower (*p* < 0.001) in comparison to the other methods (Table 5).

### 3.3. Rare Complications

Moreover, our exploration highlighted several rare complications documented within the scope of endoscopic spine surgery. These included cases of hematoma in the iliopsoas, incorrect positioning of the endoscopic access system, instrument entrapment in the working channel, segmental artery injury, pneumothorax, pseudomeningocele with nerve root entrapment, discal pseudocyst, and arachnoid cyst (Table 6).

### 3.4. Case 1: Transient Cauda Equina Syndrome Due to Subdural Hematoma

A 36-year-old male was scheduled for surgery due to a left-sided L5/S1 disc extrusion (Figure 2). The patient had undergone surgery at this level on the opposite side 10 years earlier. A left-sided interlaminar approach was chosen. During the procedure, difficulties were encountered in mobilizing neural structures while rotating the working channel. However, after several attempts at mobilization, the extrusion was visualized and removed in several small fragments. Upon awakening, the patient reported numbness in the perianal region and difficulty urinating. Examination revealed a slight decrease in rectal sphincter tone and subtle weakness in the dorsal flexion of both feet. A follow-up MRI showed shrinkage of the dural sac at the operated level (Figure 3). A decision was made for open revision surgery, performed several hours after the initial procedure. During the revision, damage to the dural sac on the right side was found, along with a small subdural hematoma located among the nerve fibers, which was not visible during the endoscopic procedure. The hematoma was removed, the sac was sutured, and it was sealed with Tachosil. Postoperatively, the patient experienced slight improvement but required further urorehabilitation. Subsequent MRIs showed gradual absorption of postoperative changes and the sealing of the dural sac. With stimulation and rehabilitation, there was gradual improvement, a return of sexual function, and sphincter efficiency. Three months later, the patient experienced a gradual recurrence of disorders, especially sexual function. An MRI revealed a small recurrent herniation. Due to the absence of pain syndrome, the patient was initially treated conservatively, but decided on revision surgery at another center a week later. Post-surgery, there was an improvement in function, but slight deterioration occurred a few days later. An MRI showed a small iatrogenic joint cyst on the left side, causing slight stenosis. The postoperative course included a superficial MSSE infection treated with antibiotics. Further MRIs showed the resolution of the changes with the absorption of the cyst (Figure 4). Ultimately, the patient healed, functioning normally with proper sexual functions, though occasionally experiencing discomfort in the perianal area.

### 3.5. Case 2: Sudden Cardiac Arrest Due to Air Embolism

A 30-year-old female patient, with no medical history and not on hormonal contraception, presented with acute bilateral sciatica due to a massive L4/L5 disc extrusion. She was scheduled for a right-sided transforaminal discectomy. The procedure involved access to and removal of a massive, sequestrated disc fragment, relieving pressure on the neural structures. During the final phase of hemostasis, several small air bubbles appeared in the fluid administered via the pump, likely due to the depletion of the fluid bag. Seconds later, a change in the ECG trace and a drop in pCO_2_ were observed. The procedure was immediately halted, the patient was turned supine, and symptoms of embolism—significant rhythm disturbances, cyanosis around the neck, and the absence of detectable pulse—prompted the initiation of indirect massage and the administration of 4 mg of adrenaline in cycles, plus 1 mg of atropine. Despite these interventions, oxygen saturation remained above 80%, and pupils were narrow and equal. Within 10 min, the patient’s own heart activity, initially arrhythmic, returned to a hemodynamically stable sinus rhythm. The patient, sedated with propofol, was transferred to the Intensive Care Unit. Upon stabilization, the patient was awakened without any neurological deficits and was moved to the neurosurgery department the following day for complete cardiological diagnostics. No other cause for the arrest was found, and no cardiovascular abnormalities were detected. The circumstances and progression suggest an occurrence of air embolism.

### 3.6. Case 3: Transient Paraparesis following Iatrogenic Vascular Fistula of the Dural Sac

A 50-year-old female patient was scheduled for endoscopic surgery to treat L5/S1 discopathy (Figure 5 and Figure 6). A posterior access was made, and the endoscope was navigated through the yellow ligament via an interlaminar window, which was sufficiently wide and required no additional drilling. Neural structures were mobilized, and the herniation was removed. Post-awakening, the patient was pain-free and mobilized with physiotherapist assistance on the same day. Upon standing, she experienced acute lumbar pain, preventing movement. A control MRI revealed an acute hematoma within the dural sac (Figure 7). Due to persistent discomfort 24 h post-surgery, an unsuccessful lumbar puncture attempt was made, followed by a decision for conservative treatment. On day 4, the patient developed complete lower limb paralysis, loss of sensation, and sphincter paralysis. A follow-up MRI showed hematoma progression to the conus medullaris level (Figure 8). Despite the dramatic progression, surgical revision was not undertaken. The following day, sensation and movement gradually returned to the limbs, and the patient was rehabilitated. By day 11, she was mobilized and discharged with functioning sphincters but persistent acute LS pain syndrome. Over the next two months, she reported LS pain, severe headaches, and episodes of unconsciousness. Subsequent MRIs showed gradual hematoma resolution (Figure 9). Approximately two months post-surgery, the patient regained full function, and acute pain subsided.

The patient had no medical comorbidities, coagulation disorders, hypertension, or medication use. The surgery occurred during the last days of menstruation. Upon case review, the surgeon and radiologist concluded that the only cause of the progressing intradural hematoma was an iatrogenic fistula between a small vessel and the dural sac interior. No dural sac damage or significant deviation from normal bleeding was noted during the procedure.

## 4. Discussion

Spinal endoscopy is increasingly acknowledged as a pivotal advancement in the management of discopathy, signaling a shift away from traditional tubular techniques and classic microdiscectomy. This evolution, highlighted over the past decade, posits endoscopy as potentially the future norm for such treatments [44]. Despite this, contemporary literature predominantly suggests that the clinical outcomes of endoscopic treatment align closely with those of conventional approaches [32,33]. Nonetheless, certain studies underscore the advantages of endoscopy, including lower complication rates, shorter hospital stays, and reduced procedure times, which collectively contribute to a faster return to professional activities and socio-economic benefits [45].

Our review corroborates the perception that endoscopic procedures generally exhibit lower complication rates. However, this does not negate the presence of complications. Notably, the variation in reported incidences of incidental durotomy across meta-analyses hints at possible underreporting [9,10,11]. While such complications are often deemed benign, our case reports illustrate that their repercussions can be profound, as seen in instances of cauda equina syndrome and iatrogenic vascular fistula of the dural sac.

A critical observation from our examination of meta-analyses and RCTs highlights a methodological concern: the aggregation of transforaminal and interlaminar endoscopic spine surgeries into a single analysis group. This approach may mask significant technical differences and risk profiles between the techniques. Domenico Compagnone et al.’s [13] systematic review is exceptional in its methodological rigor, distinctly analyzing the transforaminal and interlaminar approaches and thus providing a nuanced perspective on the spectrum of complications unique to each.

Despite these distinctions, Compagnone’s findings reveal that the complication risks for both approaches are comparably low. This challenges the premise that technical and procedural differences significantly impact safety and complication rates, suggesting instead that the surgeon’s expertise and the meticulous execution of the procedure are paramount. This observation supports the broader adoption of endoscopic spine surgery, emphasizing the importance of comprehensive training and adherence to best practices.

In our analysis, the most common complications identified were sensory disturbances, incomplete herniation removal, durotomy, nerve root damage, hematomas, discitis, and instability. It is noteworthy that none of these complications appear to be exclusively characteristic of endoscopic procedures, as the same spectrum of complications occurs in microscopic surgeries. This observation suggests that the inherent risks associated with spinal surgeries transcend the specific methodologies employed, be they endoscopic or microscopic approaches.

However, when considering casuistic complications such as hematoma in the psoas muscle, malposition of the working channel, entrapment of instruments, and injury to the radicular artery, these appear to be unique to endoscopic procedures. The literature review, including the work of Zhou et al. [35], supports this distinction, indicating that such complications have not been commonly reported in microdiscectomy procedures. This specificity underscores the unique challenges and potential risks associated with endoscopic spine surgery, highlighting the importance of specialized training and awareness of these less common but possible complications.

Despite the common warnings during endoscopic courses about the risk of organ damage, such as kidney injuries during transforaminal access, our literature search found no reports of such damage occurring in the context of endoscopic spine surgeries. This could either indicate the rarity of such events or the effectiveness of the precautionary techniques employed by surgeons skilled in endoscopic procedures. However, we did come across a single case in an RCT by Yoon et al. [22] that reported a complication involving bowel damage, the circumstances of which were not fully elucidated. It is known, however, that the patient was treated conservatively and recovered.

### 4.1. Learning Curve Consideration

The meta-analysis by Rui Shi et al. [17] brings attention to the learning curve’s impact on complication risks, highlighting variances in procedure durations that reflect differences in surgeons’ skill levels. The advancement of endoscopic systems, characterized by the introduction of new tools and improved visualization technologies, underscores the importance of ongoing training and education.

The steep learning curve associated with endoscopic spine surgery, particularly noted in its slower adoption in European countries compared to Asia, underscores the challenges inherent in mastering these techniques [31]. This learning curve significantly influences complication rates and emphasizes the need for enhanced training and experience [17].

The initial exposure to spinal endoscopy during medical residency is limited, posing challenges in integrating these techniques into routine practice. The complexity of these procedures, coupled with the cost of necessary instrumentation, presents additional barriers, especially in resource-limited settings.

Furthermore, the risk of radiation exposure during transforaminal procedures highlights the need for preventive measures [46]. Despite these challenges, endoscopic surgery remains a favorable option for specific cases, underscoring the need for a balanced approach to spinal surgery.

Kotheeranurak et al.’s [47] proposal for selecting the appropriate endoscopic technique based on the pathology’s location and characteristics further refines our discussion, emphasizing the importance of technique-specific understanding in optimizing patient outcomes and minimizing complications.

### 4.2. Case Report Reflection

Case 1: Incidents of postoperative cauda equina syndrome are documented in the context of traditional surgical approaches. In our literature review, we identified a singular case of postoperative cauda equina syndrome, which, however, pertained to a patient following endoscopic intervertebral stabilization. This incident was associated with the migration of bone material into the dural sac [43]. Our case suggests that previous surgery, leading to scarring and adhesions, and anatomical challenges such as steep and large facet joints, might have contributed to the complication by complicating root mobilization and endoscope trajectory.Case 2: Air embolism during spinal surgeries is noted primarily in extensive procedures like scoliosis correction [48], according to our literature search. However, we also found a single case description from the 1970s following a discectomy operation [49]. This indicates that while the complication is more commonly linked to major surgeries, it can also occur in less extensive, traditional procedures. No endoscopic cases were found in our literature search. Nevertheless, the use of irrigation pumps instead of gravity flow in endoscopic surgeries can introduce air, potentially leading to embolism. Observations of air bubbles in the endoscopic view underscore the risk, highlighting the need for careful fluid management during these procedures.Case 3: Our review did not uncover any literature on intradural hematomas following endoscopic surgery. While there are isolated reports of such hematomas in other contexts [50,51], the dramatic progression and conservative management decision in our case are unique and underscore the need for cautious postoperative monitoring and possibly early intervention in similar future cases.

In essence, while endoscopic spine surgery heralds a new era in minimally invasive procedures, its successful implementation and outcomes heavily depend on overcoming the steep learning curve and understanding the specific limitations and complications associated with these techniques. Further advancements in instrumentation and training programs are essential for broadening the applicability and effectiveness of endoscopic spine surgery, ultimately enhancing patient care and surgical outcomes in the domain of spine surgery.

Furthermore, it is crucial to acknowledge that the fundamental goal of our article was not to prove whether endoscopic spine surgery is better or safer than classical surgical solutions. Our primary intention was to explore the literature for casuistic complications in endoscopy akin to those we have illustrated through case descriptions. This focus is paramount amid the swift popularization of the endoscopic method and its application in performing procedures in an ambulatory setting. By highlighting that such complications are possible and indeed occur, we aim to signal to the medical community the importance of being cognizant of these potential outcomes. This is especially relevant as the endoscopic approach gains traction, emphasizing the need for ongoing education, proficiency development, and heightened awareness of possible complications among practitioners. This understanding underscores the necessity of a balanced and well-considered approach to adopting new surgical techniques, ensuring that patient safety and care remain at the forefront of surgical innovation.

## 5. Conclusions

Our study underscores endoscopic spine surgery’s less invasive yet effective nature, despite potential complications that demand high surgical acumen and training. The technique’s evolving role in outpatient care for spinal disorders necessitates a focus on complication management.

Our key findings indicate that endoscopic surgery typically exhibits a lower rate of complications. However, a heightened awareness and preparedness for both typical and atypical complications are crucial. The notable learning curve associated with these techniques underscores the importance of ongoing education and the refinement of surgical skills.

In conclusion, endoscopic spine surgery presents a promising pathway for minimally invasive interventions, one that is dependent on a thorough comprehension of its complexities and a dedication to continuous surgical skill and patient care practice improvements. Our findings aim to contribute towards enhancing safety and outcomes within this evolving surgical field, promoting the judicious and informed application of endoscopic spine surgery techniques to optimize patient care.

## 6. Limitations and Strengths

This study emphasizes the importance of recognizing complications in endoscopic spinal surgery, showcasing a broad spectrum of risks. A significant strength lies in our detailed focus on specific complications, offering insights beyond some existing analyses. However, the variability in reporting standards, a notable limitation, can obscure true complication incidences, potentially introducing bias. Additionally, the concentration of endoscopic procedures among specialized surgeons might not fully represent the broader surgical community’s experience, suggesting a selection bias in reported outcomes. This disparity underscores the urgent need for standardized complication reporting to enhance research clarity and support more informed clinical decision-making.

## Figures and Tables

**Figure 1 jcm-13-01765-f001:**
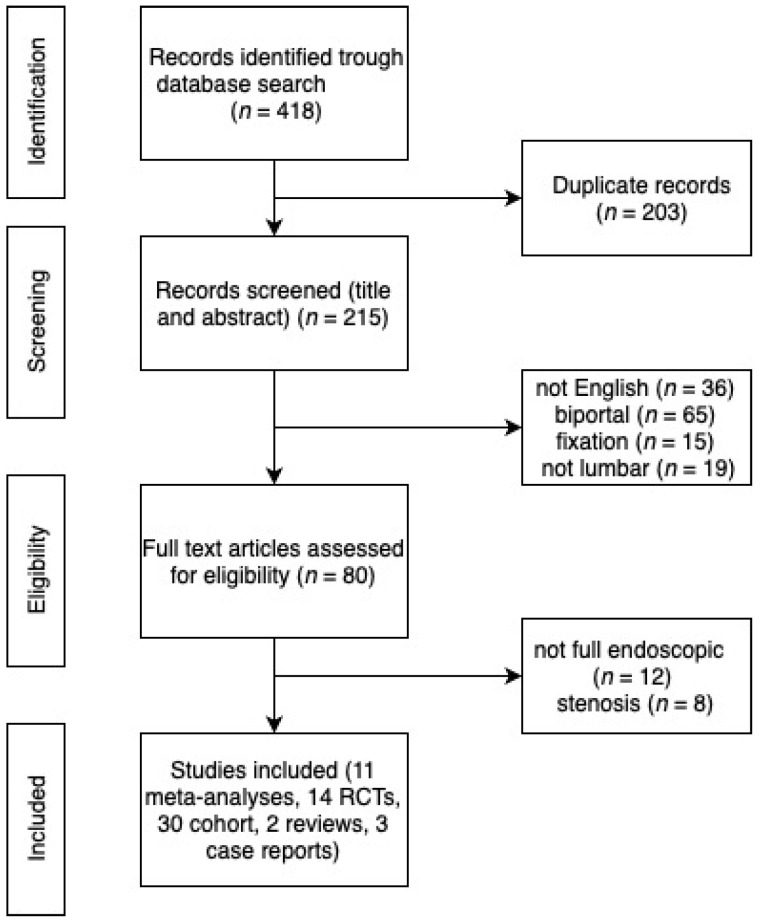
Flowchart of the Study Selection Process.

**Figure 2 jcm-13-01765-f002:**
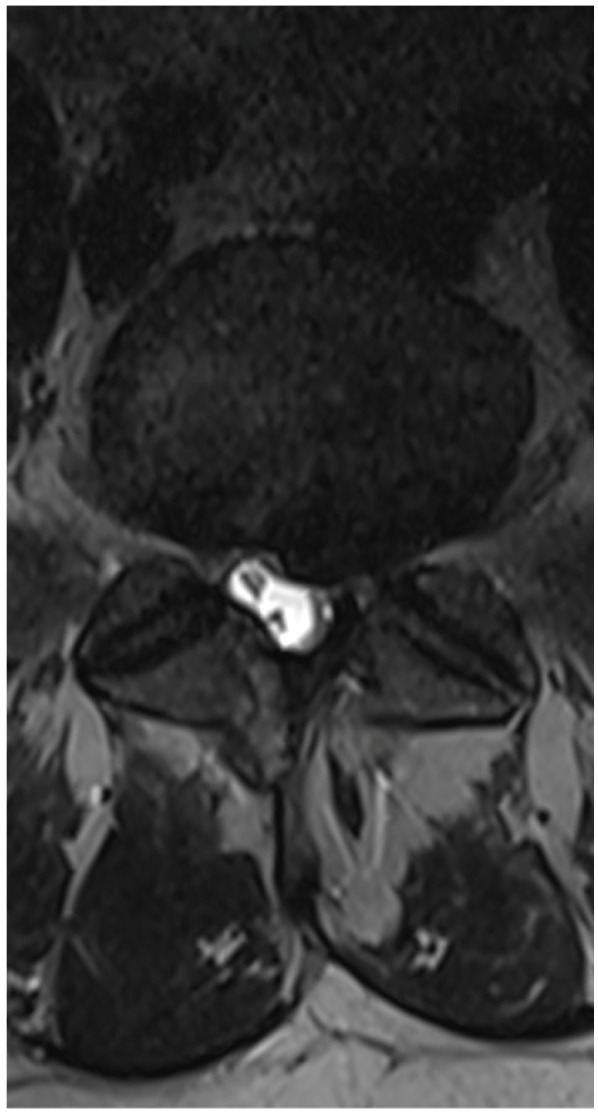
Preoperative T2 axial MRI on L5/S1 level.

**Figure 3 jcm-13-01765-f003:**
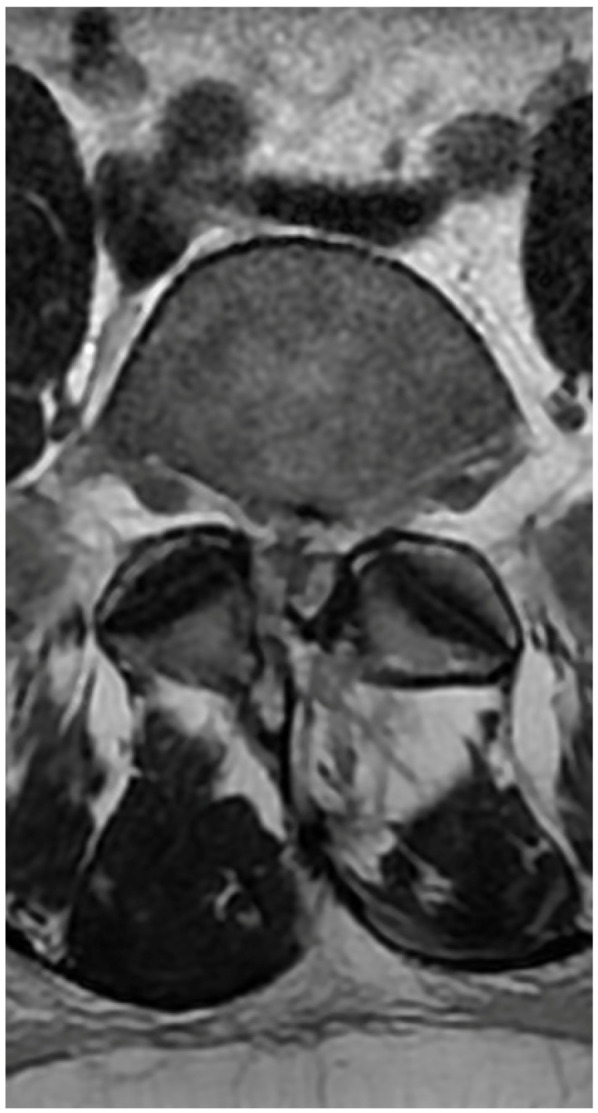
Sagittal T2 MRI 24 h after the surgery.

**Figure 4 jcm-13-01765-f004:**
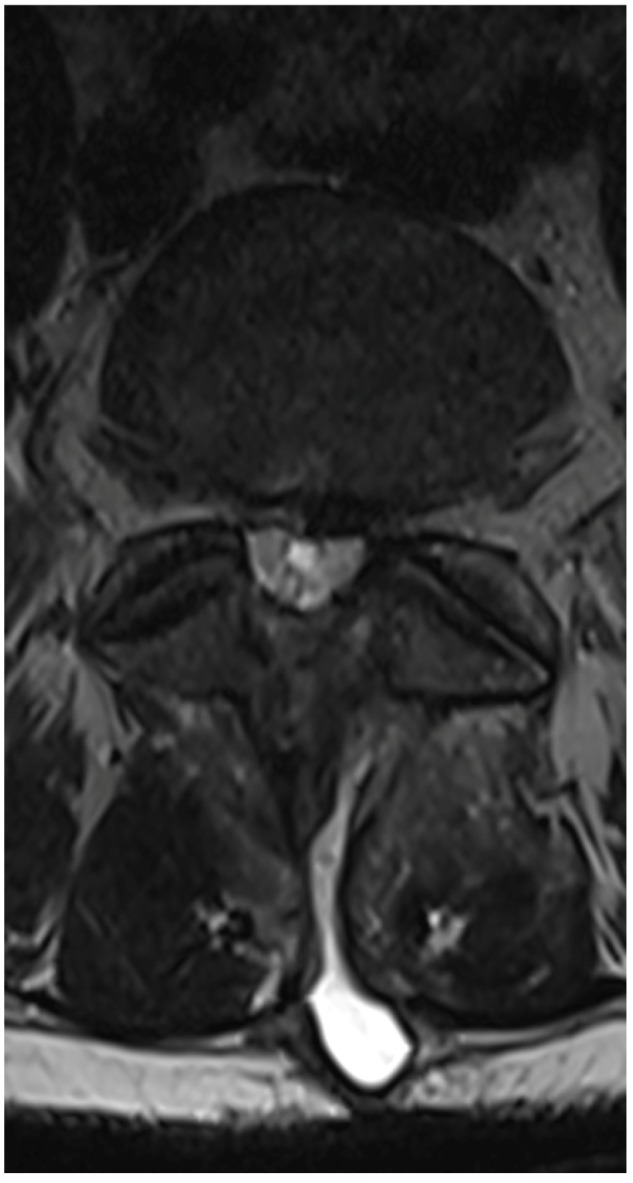
Sagittal T2 MRI after third surgery.

**Figure 5 jcm-13-01765-f005:**
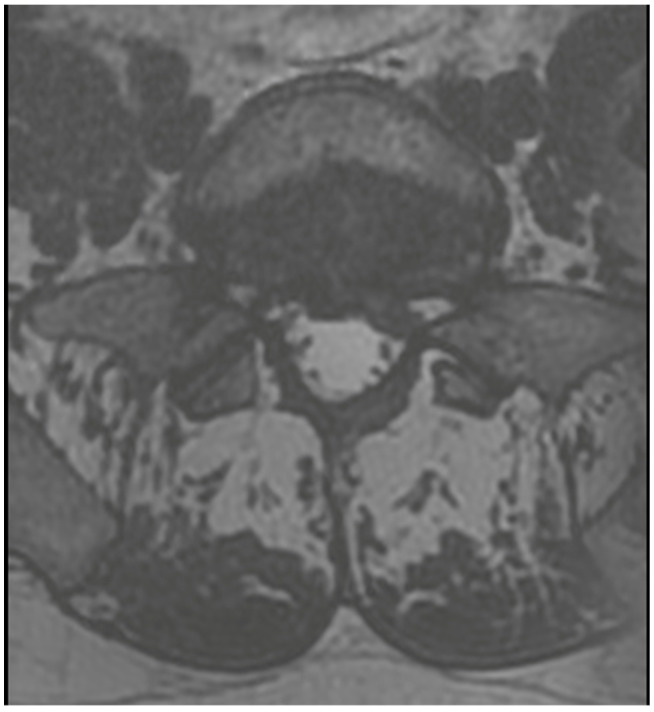
Preoperative T2 axial MRI on the L5/S1 level.

**Figure 6 jcm-13-01765-f006:**
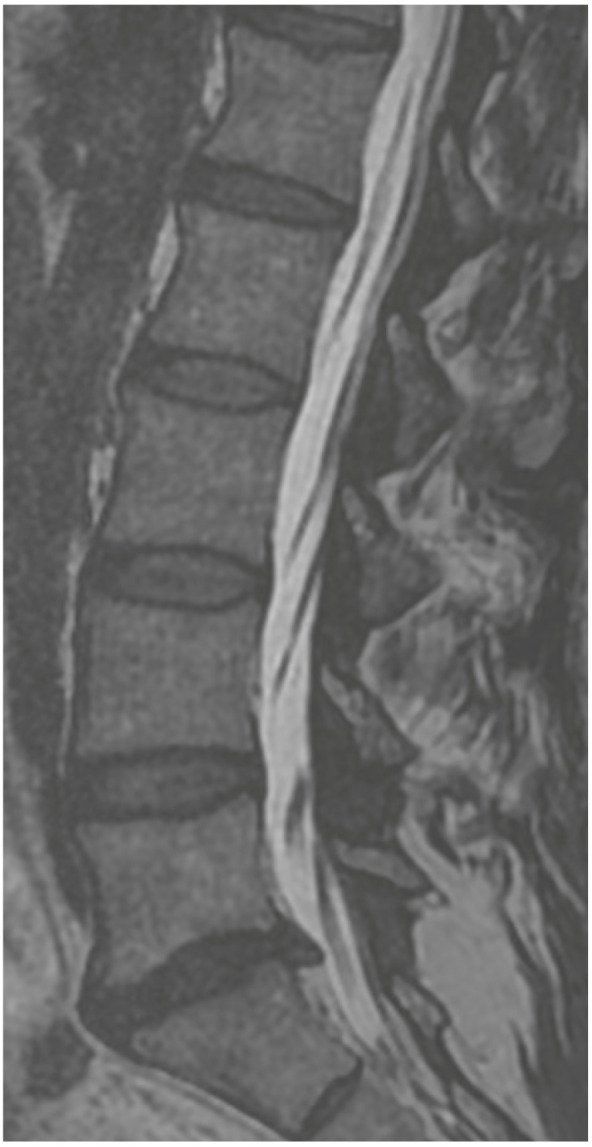
Preoperative T2 sagittal MRI.

**Figure 7 jcm-13-01765-f007:**
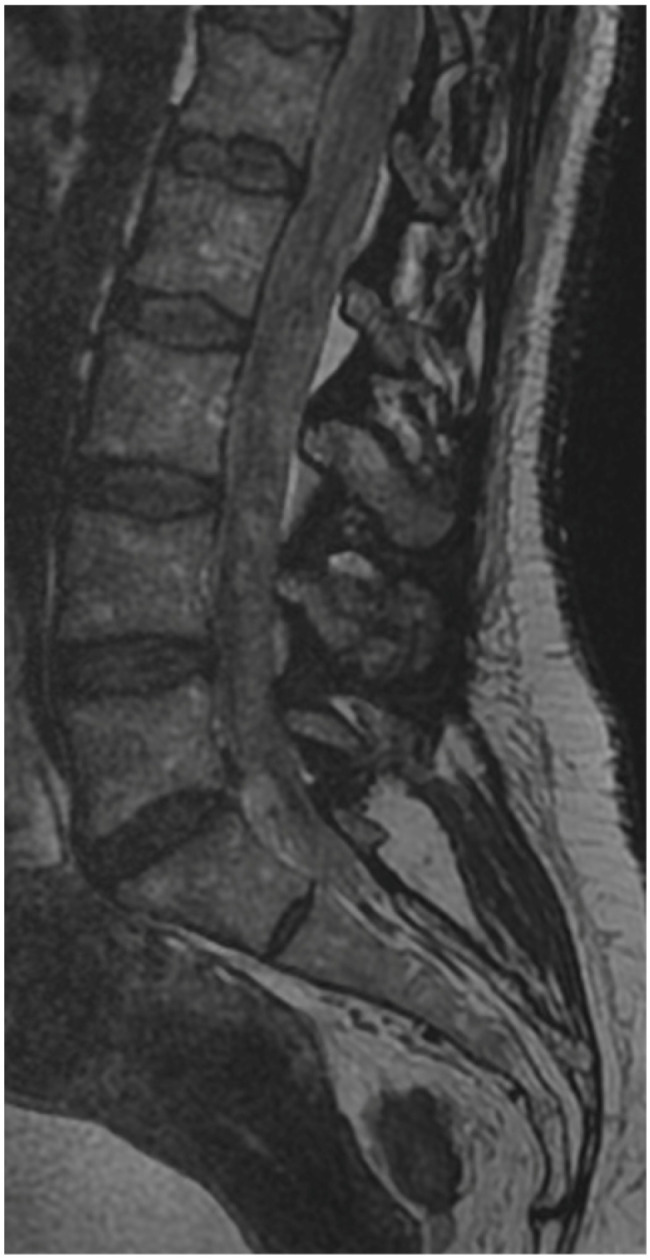
Sagittal T2 MRI 24 h after the surgery.

**Figure 8 jcm-13-01765-f008:**
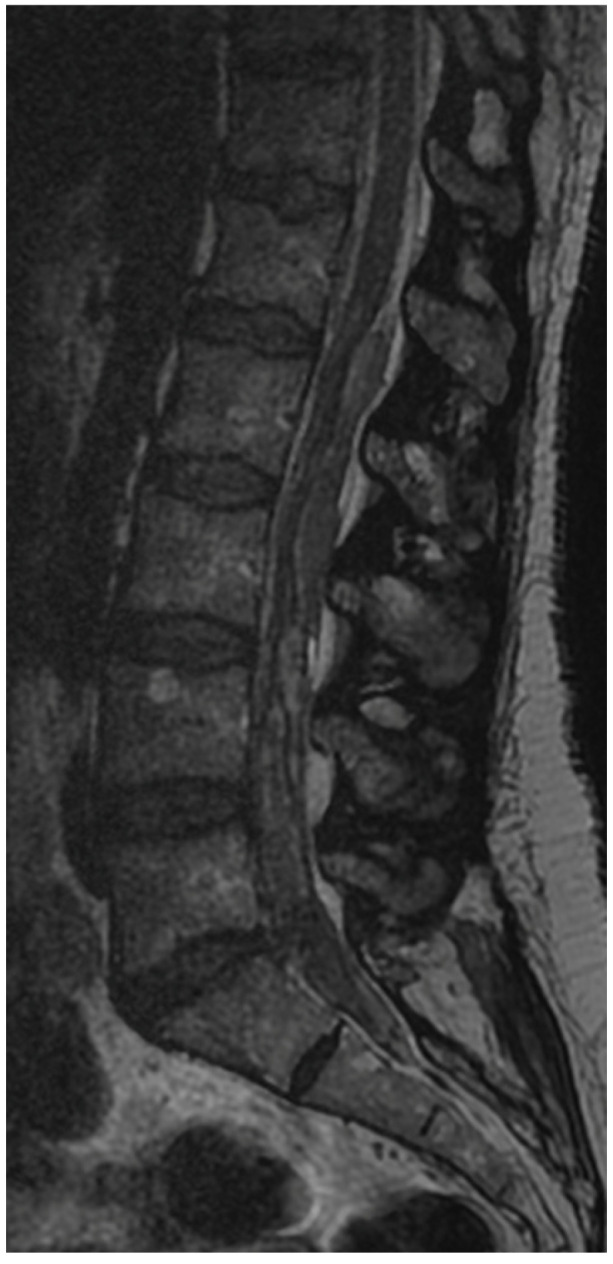
Sagittal T2 MRI 96 h after the surgery.

**Figure 9 jcm-13-01765-f009:**
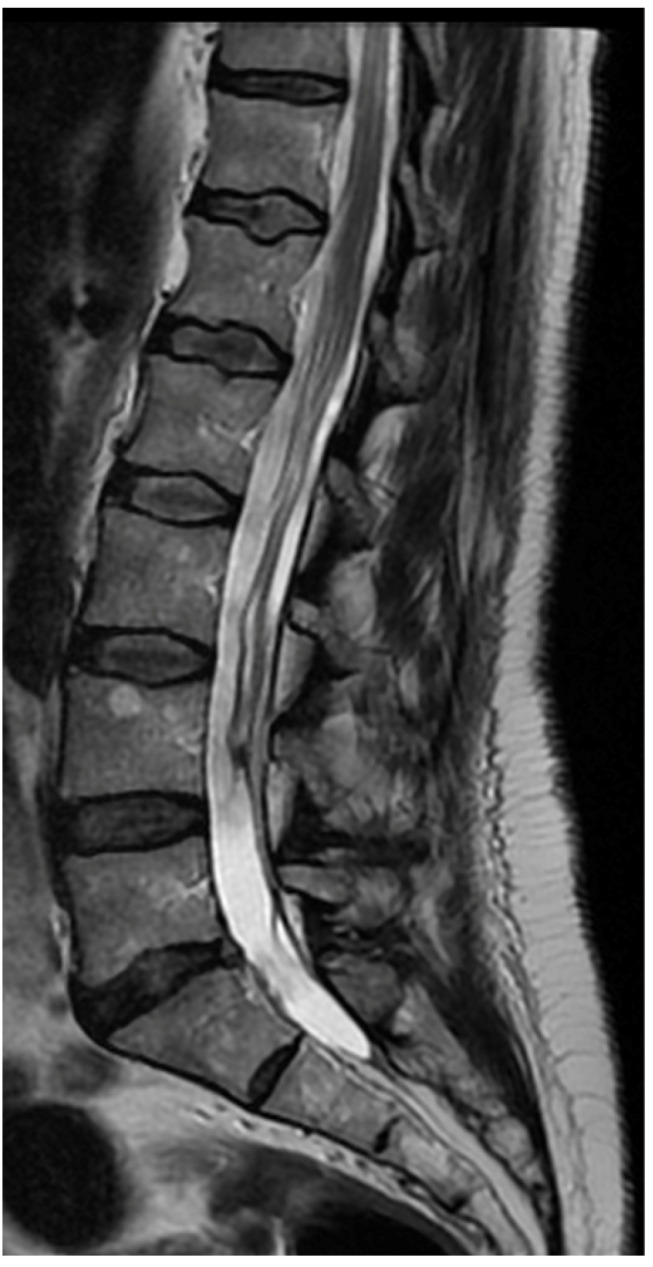
Sagittal T2 MRI 2 months after the surgery.

**Table 1 jcm-13-01765-t001:** Complication rates (%) in transforaminal endoscopic spine surgery (TELD), interlaminar endoscopic spine surgery (IELD), and micro lumbar discectomy (MLD) in meta-analyses.

Author	Journal	TELD	IELD	MLD
Compagnone et al. (2023) [13]	*European Spine Journal*	5.2	3.1	N/A
Li et al. (2022) [9]	*Global Spine Journal*	4.3		14.6
Zhao et al. (2022) [14]	*Journal of Clinical Medicine*	No significant differences		
Yang et al. (2022) [10]	*World Neurosurgery*	5.5 (RCT meta-analysis) 6.4 (Cohort meta-analysis)		10.4 (RCT meta-analysis) 4.0 (Cohort meta-analysis)
Zhang et al. (2022) [15]	*Frontiers in surgery*	No significant differences		
Bai et al. (2021) [16]	*Medicine*	6.79		6.36
Chen et al. (2020) [11]	*European Spine Journal*	5.8		16.8
Shi et al. (2019) [17]	*International Orthopaedics*	6.8 (54/778)		7.61 (57/749)
Kim et al. (2018) [18]	*Hindawi* *BioMed Research International*	No significant differences		
Qin et al. (2018) [19]	*World Neurosurgery*	3.6		3.53
Ruan et al. (2016) [20]	*International Journal of Surgery*	4.69		2.33
Pooled Risk		5.34 ± 1.32		8.2 ± 3.95

**Table 2 jcm-13-01765-t002:** Common complication rates (%) in endoscopic procedures in meta-analyses.

	Compagnone et al. (2023) [13]		Shi et al. (2019) [17]	Qin et al. (2018) [19]	Yang et al. (2022) [10]		Zhou et al. (2018) [14]	Pooled Risk
	IELD	TELD			RCT	COHORT		
hematoma	0.06	0.04	N/A	0	0	0.1	N/A	0.05 ± 0.03
residual disc	0.3	1.0	4.25	N/A	0.7	3.1	1.4	1.79 ± 1.49
neurologic deficit	0.3	0.3	N/A	N/A	0	0.1	N/A	0.23 ± 0.2
dysesthesia	1.3	1.5		1.7	4.1	2.6	N/A	2.24 ± 0.94
dural tear	0.9	0.7	0.68	0.76	1.0	0.7	0.9	0.81 ± 0.11
nerve root injury	N/A	N/A	N/A	0.38	0	0.6	1.2	0.55 ± 0.53
discitis	N/A	N/A	N/A	0.38	0	N/A	N/A	N/A
instability	0.1	0.2	N/A	N/A	N/A	0	N/A	N/A
infection	N/A	N/A	N/A	0	0	0	N/A	N/A
reoperation	N/A	N/A	4.03	N/A	4.8	7.6	N/A	5.48 ± 2.08

**Table 3 jcm-13-01765-t003:** Complication rates in transforaminal endoscopic spine surgery (TELD), interlaminar endoscopic spine surgery (IELD), percutaneus endoscopic spine surgery (PELD), and micro lumbar discectomy (MLD) in RCTs.

Author	Type	Age	Quantity (n)	Dural Tear	Neural Injury	Transient Dysesthesia	Persisted Pain	Hematoma	Motor Weakness	Post Operative Urinary Retention	Wound Heeling/INFECTION	Residue/Recurrence	Total No. of Complications	Reoperation
Rueten et al. (2008) [21]	TELD	43	41	0	0	1	0	0	0	0	0	3	4	3
	IELD	43	59	0	0	2	0	0	0	0	0	4	6	4
	MLD	43	100	0	0	5	0	2	0	3	2	5	17	5
Yoon et al. (2012) [22]	PELD	45.88	37	1	0	0	0	0	0	0	0	3	3	0
	MLD	56.46	35	1	0	0	0	0	0	0	0	2	2	0
Lei Pan et al. (2014) [23]	PELD	N/A	10	0	0	1	0	0	0	0	0	0	1	0
	MLD	N/A	10	0	0	0	0	0	0	0	0	0	0	0
Gibson et al. (2016) [24]	TELD	42 ± 9	70	2	0	4	0	0	0	0	0	3	9	5
	MLD	39 ± 9	70	0	0	0	0	0	1	0	0	0	1	2
Zhimin Pan et al. (2016) [25]	TELD	39.5	48	0	0	3	0	0	0	0	0	0	3	0
	MLD	42.8	58	2	0	3	0	0	4	4	0	0	13	0
Liu et al. (2018) [26]	TELD	36.2 ± 5.9	60	0	0	0	2	0	0	0	0	3	5	0
	MED	33.1 ± 6.2	89	3	0	0	0	0	0	0	1	2	6	0
	MLD	34.0 ± 3.8	105	2	0	0	0	0	0	0	3	0	5	0
Chen et al. (2018) [27]	TELD	40.9 ± 11.9	80	1	3	2	0	0	0	0	0	5	11	5
	MED	41.0 ± 10.8	73	1	0	7	0	0	0	0	1	3	12	3
Chen et al. (2019) [28]	TELD	40.9 ± 11.9	119	1	3	2	0	0	0	0	0	6/4	16	10
	MLD	41.0 ± 10.8	122	3	1	9	0	0	0	0	1	0/5	19	5
Meyer et al. (2020) [29]	PELD	47.2 ± 10.6	23	0	0	0	0	0	0	0	0	3	3	3
	MLD	45.2 ± 10.6	24	1	0	0	0	0	0	0	1	2	4	3
Chen et al. (2022) [30]	TELD	40.9 ± 11.9	97	N/A	N/A	N/A	N/A	N/A	N/A	N/A	N/A	7	N/A	5
	MLD	41.0 ± 10.8	97	N/A	N/A	N/A	N/A	N/A	N/A	N/A	N/A	7	N/A	7
Gadjradj et al. (2022) [31]	TELD	45.3 (12.4)	179	0	0	2	0	0	0	0	0	N/A	2	9
	MLD	45.7 (11.3)	249	2	0	0	0	1	0	1	3	N/A	7	14
Tang et al. (2023) [32]	PELD	40.44 ± 8.23	25	0	0	1	0	0	0	0	0	1	2	0
	MLD	37.80 ± 9.35	25	0	0	1	0	0	0	0	0	2	3	1
Sharma et al. (2024) [33]	TELD	35 ± 15.78	220	1	2	2	1	0	0	0	1	3	10	4
	MLD	38 ± 17.49	220	2	2	1	2	1	0	0	5	1	14	2
Total (%)	TELD		914	0.55	0.88	1.75	0.33	0	0	0	0.11	3.72	**7.33**	4.49
	IELD		59	0	0	3.39	0	0	0	0	0	6.78	**10.17**	6.77
	PELD		1009	0.50	0.79	1.78	0.30	0	0	0	0.10	3.77	**7.23**	4.52
	MLD		1115	1.17	0.27	1.7	0.18	0.36	0.45	0.72	1.35	2.15	**8.34**	3.5

**Table 4 jcm-13-01765-t004:** Comparative analysis of complication rates among lumbar discectomy techniques.

Procedures	*p* Value	Risk Ratio (RR)	Statistical Significance
TELD vs. IELD	*p* < 0.0001	RR = 0.72	TELD safer
TELD vs. MLD	*p* = 0.18	RR = 0.98	no difference
IELD vs. MLD	*p* = 0.04	RR = 1.27	MLD safer
MLD vs. PELD	*p* = 0.0092	RR = 1.06	PELD safer

**Table 5 jcm-13-01765-t005:** Comparative analysis of reoperation rates among lumbar discectomy techniques.

Procedures	*p* Value	Risk Ratio (RR)	Statistical Significance
TELD vs. IELD	*p* < 0.0001	RR = 0.66	TELD less reoperations
TELD vs. MLD	*p* < 0.0001	RR = 1.28	MLD less reoperations
IELD vs. MLD	*p* < 0.0001	RR = 1.93	MLD less reoperations
MLD vs. PELD	*p* < 0.0001	RR = 0.77	MLD less reoperations

**Table 6 jcm-13-01765-t006:** Rare complication rates in endoscopic procedures in RCTs and cohort studies.

Rare Complication Type	Author
psoas muscle hematoma	Ahn et al. (2009) [34]
working channel malposition	Zhou et al. (2018) [35]
instrument entrapment	Zhou et al. (2018) [35] Zhu et al. (2017) [36]
radicular artery injury	Zhou et al. (2018) [35] Wang Y. et al. (2018) [37]
negative pressure pulmonary oedema	Chen G. et al. (2018) [38]
pseudomeningocele with nerve root entrapment	Shu W. et al. (2020) [39]
discal pseudocyst	Li J et al. (2020) [40]
arachnoid cyst	Lou X et al. (2023) [41]
seizure	Zhang et al. (2022) [42]
cauda equina syndrome	Yang et al. [43]
bowel injury	Yoon et al. [22]

## Data Availability

The original contributions presented in the study are included in the article, further inquiries can be directed to the corresponding author.

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
