# Peer review of "Extremely Rare Complications in Uniportal Spinal Endoscopy: A Systematic Review with Unique Case Analyses"

_jcm, 2024, doi:10.3390/jcm13061765_

Round 1

Reviewer 1 Report

Comments and Suggestions for Authors

Authors have conducted a systematic review on uniportal spinal endoscopy with a focus on its specific complication profile. They focused in discectomies instead of more complex spine surgeries. They reported that complication rates have been reported lower with endoscopic procedures compared to microdiscectomies. Yet, endoscopic surgeries carry a risk of unique, rare complications not encountered in conventional surgeries.

This topic is highly relevant considering a new technique that is emerging with hype and conveys increased costs.

The strength of this study was reporting on rare complications unique to endoscopic surgery. Depiction of cases was informative and interesting. Although authors claimed to have followed PRISMA guidelines, they did not assess risk of bias in included studies. For example, references 6,7,8 were retrospective comparisons. It is likely that endoscopic procedures have been performed by more experienced surgeons and conventional microdiscectomies by larger and more heterogeneous group of surgeons thus making the comparison flawed favoring endoscopies. This limitation should be highlighted in the text, although it does not excessively strain this study as authors focused on reporting specific complications.

In the results section and Tables 4 and 5; significant findings were reported only by p values. It would be more informative for readers to report odds ratios or risk ratios for complications btw groups.

Lastly, I suggest making conclusions paragraph more succinct.

Author Response

We appreciate the insightful comments and constructive criticisms provided by the reviewer. Addressing the concerns raised:

  1. Risk of Bias Assessment: We acknowledge the omission of a detailed risk of bias assessment for the included studies, primarily because our article's core focus was on identifying specific, unique complications of uniportal spinal endoscopy. The exploration of common complications served more as a contextual background. It is pertinent to note that the majority of existing publications, including retrospective comparisons and even Randomized Controlled Trials (RCTs), exhibit a considerable risk of bias. This observation aligns with the reviewer's accurate remarks regarding the potential disparities in surgeon experience between endoscopic and microdiscectomy procedures. To address this critical issue, we have incorporated an additional limitations paragraph, explicitly highlighting the concern regarding the risk of bias in the literature reviewed and its implications on the comparability of complication rates between endoscopic surgeries and microdiscectomies.

  2. Revision of Tables 4 and 5: In response to the reviewer’s suggestion for a more informative presentation of significant findings, we have revised Tables 4 and 5 to include Risk Ratios (RR) alongside the p-values. This amendment aims to provide a clearer understanding of the comparative risk of complications between groups, enhancing the article's informative value for our readers.

  3. Conclusions Paragraph: In line with the recommendation for conciseness, we have meticulously revised and shortened the conclusions paragraph. This revision is intended to succinctly encapsulate the study's key findings and implications, ensuring clarity and directness in communicating our research outcomes.

We believe these amendments significantly strengthen the manuscript by addressing the limitations initially overlooked and enhancing the clarity and comprehensiveness of our findings. We are grateful for the opportunity to refine our work based on the reviewer's valuable feedback.

Reviewer 2 Report

Comments and Suggestions for Authors

I appreciate the opportunity to review the manuscript, entitled “Extremely Rare Complications in Uniportal Spinal Endoscopy: A Systematic Review with Unique Case Analyses.”

The article provides a valuable contribution to the literature on endoscopic spine surgery by highlighting the spectrum of complications and emphasizing the need for surgeon education and skill development. I have the following questions and suggestions:

1. In the "Complication rates" section on page 4, lines 147-155, the article mentions that nine studies calculated the incidence rates of complications, and only two studies differentiated between the transforaminal and interlaminar approaches. However, in Table 1, only eight studies calculated the incidence rates of complications, and only one study differentiated between the transforaminal and interlaminar approaches. Additionally, three meta-analyses indicated a significant difference in the frequency of complications favoring endoscopy; please indicate in Table 1 which three studies these are. If possible, conduct a pooled analysis of the data from the literature to calculate the overall incidence rate of complications.

2. In the "Common Complication" section on page 5, lines 166-169, how is the incidence rate of common complications calculated? If possible, summarize and indicate the incidence rates of different complications in Table 2.

3. The text does not refer to Table 3; please double-check.

4. In the discussion section on page 12, lines 349-350, the order of common complications is inconsistent with the description in the "Common Complication" section on page 5, lines 166-169; please verify.

Comments on the Quality of English Language

There is no major issues with English language writing.

Author Response

We are grateful for your insightful feedback and constructive suggestions on our manuscript. Below are our responses and the actions we have taken to address your comments:

  1. Discrepancy in "Complication rates" section and Table 1: We sincerely apologize for the oversight in reporting the number of studies calculating the incidence rates of complications and differentiating between the transforaminal and interlaminar approaches. This was an editorial error, and we have since corrected the text to accurately reflect the number of studies mentioned. Furthermore, we have updated text to include references to the three meta-analyses that demonstrated a significant difference in the frequency of complications favoring endoscopy. In response to your suggestion, we conducted a pooled analysis to calculate an overall incidence rate of complications and each complication when possible.

  2. Calculation of Incidence Rate in "Common Complication" section: We have clarified the methodology for calculating the complication rate of common complications in the materials and methods section. The complication rate was calculated as the number of each complications divided by the total number of procedures performed in a given group, multiplied by 100%. 

  3. Reference to Table 3: Upon re-examination, we realized that Table 3 was indeed not referenced appropriately within the text. This error has been corrected, ensuring that Table 3 is now accurately cited and contributes to the discussion as intended.

  4. Inconsistency in Order of Common Complications: We acknowledge the inconsistency in the order of common complications between the discussion section and the "Common Complication" section. This has been rectified, and we have standardized the sequence of complications in both sections to ensure consistency throughout the manuscript.

We appreciate the opportunity to improve our manuscript based on your valuable feedback. We believe that these amendments enhance the clarity and accuracy of our study, contributing to a more comprehensive understanding of the complications associated with uniportal spinal endoscopy.

Round 2

Reviewer 2 Report

Comments and Suggestions for Authors

I enjoy reading the revised manuscript. The authors addressed most comments raised by other reviewers and me. In my opinion, the manuscript has improved and can be considered for publication. The authors also provided a very clear overview of the comments and how they addressed them.